# Abundant genetic variation is retained in many laboratory schistosome populations

**Kathrin S. Jutzeler**[1,2]*, **Roy N. Platt**[3], **Robbie Diaz**[1], **Madison Morales**[1], **Julie Dunning Hotopp**[4], **Winka Le Clec'h**[1], **Frédéric D. Chevalier**[1], **Timothy J. C. Anderson**[3]*

**1** Host Parasite Interaction Program, Texas Biomedical Research Institute, San Antonio, Texas, United States of America, **2** UT Health, Microbiology, Immunology and Molecular Genetics, San Antonio, Texas, United States of America, **3** Disease Intervention and Prevention Program, Texas Biomedical Research Institute, San Antonio, Texas, United States of America, **4** Institute for Genome Sciences, University of Maryland School of Medicine, Baltimore, Maryland, United States of America

* ksjutzeler@gmail.com (KSJ); tanderso@txbiomed.org (TJCA)

## Abstract

Schistosomes are obligately sexual blood flukes that can be maintained in the laboratory using freshwater snails as intermediate and rodents as definitive hosts. The genetic composition of laboratory schistosome populations is poorly understood: whether genetic variation has been purged due to serial inbreeding or retained is unclear. We sequenced 19 – 24 parasites from each of five laboratory *Schistosoma mansoni* populations and compared their genomes with published exome data from four *S. mansoni* field populations. We found abundant genomic variation (0.897 – 1.22 million variants) within laboratory populations: these carried on average 62% ($\pi$ = 1.52e-04 – 7.15e-04) less nucleotide diversity than the four field parasite populations ($\pi$ = 9.06e-03 – 2.24e-03). However, the pattern of variation was very different in laboratory and field populations. Tajima's D was positive in all laboratory populations (except SmBRE), indicative of recent population bottlenecks, but negative in all field populations. Current effective population size estimates of laboratory populations were lower (2 – 258) compared to field populations (3,174 – infinity). The distance between markers at which linkage disequilibrium (LD) decayed to 0.5 was longer in laboratory populations (59 bp – 271 kb) compared to field populations (9 bp – 17.1 kb). SmBRE was the least variable laboratory population; this parasite also shows low fitness across the lifecycle, consistent with inbreeding depression. The abundant genetic variation present in most laboratory schistosome populations has several important implications: (i) measurement of parasite phenotypes, such as drug resistance, using laboratory parasite populations will determine average values and underestimate trait variation; (ii) genome-wide association studies (GWAS) can be conducted in laboratory schistosome populations by measuring phenotypes and genotypes of individual worms; (iii) genetic drift may lead to divergence in schistosome populations maintained in different laboratories. We conclude that the abundant

**Data availability statement:** o The sequencing data generated for this project are available on Sequence Read Archive (SRA) under BioProject PRJNA1074697 (SmEG, SmOR, SmLE, SmNMRI) and PRJNA1170908 (SmBRE). Exome sequences from field samples have previously been published by Platt et al. [19] and are available on SRA under BioProjects PRJNA743359 (Brazil) and PRJNA560070 (Niger, Senegal, and Tanzania). All shell and R scripts written for this project are available at https://github.com/kathrinsjutzeler/sm_single_gt and Zenodo https://doi.org/10.5281/zenodo.10672478.

**Funding:** This research was supported by a Graduate Research in Immunology Program training grant NIH T32 AI138944 (KSJ), and NIH R21 AI171601-02 (FDC, WL), R01 AI133749, R01 AI166049 (TJCA), and was conducted in facilities constructed with support from Research Facilities Improvement Program grant C06 RR013556 from the National Center for Research Resources. SNPRC research at Texas Biomedical Research Institute is supported by grant P51 OD011133 from the Office of Research Infrastructure Programs, NIH. The funders had no role in study design, data collection and analysis, decision to publish, or preparation of the manuscript.

**Competing interests:** The authors have declared that no competing interests exist.

genetic variation retained within many laboratory schistosome populations can provide valuable, untapped opportunities for schistosome research.

## Author summary

Most protozoan, bacterial or viral pathogens are maintained as asexual lineages in the laboratory. This simplifies research, because the single pathogen genotypes can be studied. In contrast, many helminth parasites have two sexes and are maintained as obligately sexual populations. These populations are often referred to as "strains" and treated akin to clonal bacteria, but levels of genetic variation retained within populations is typically unknown. The blood fluke *Schistosoma mansoni* (Phylum: Platyhelminthes; *Class: Trematoda)* is a commonly used laboratory helminth. We sequenced individual parasites from five different laboratory populations of *Schistosoma mansoni* maintained for up to 80 years in the laboratory to directly measure genetic variation. We found abundant variation in four of five laboratory populations studied, with up to 1.22 million variants per population; variation in the five laboratory populations was 62% less than that present in the four field collected populations. The abundant variation retained provides both opportunities and issues for researchers. On the positive side, genetically variable laboratory populations can be used to examine the genetic basis of important parasite traits, such as drug resistance or host specificity, using genome-wide association. On the negative side, measurement of parasite traits in worm populations will determine average values and underestimate trait variation for individual parasites. Furthermore, parasite populations may diverge due to genetic drift, resulting in poor repeatability between studies.

## Introduction

Many viral, bacterial and protozoan pathogens can be cloned and maintained as asexual lineages in the laboratory. This has many advantages for research because experimental infections can be established using genetically homogeneous pathogens, and differences in biomedically important pathogen traits can be directly attributed to genetic differences between pathogen clones. In contrast, most helminth parasites (e.g., *Trichuris spp*, *Heligosomoides spp*, filarial nematodes) used in biomedical research are obligately sexual. For example, our laboratory focuses on the blood fluke, *Schistosoma mansoni*, which has separate sexes (males are ZZ; females are ZW): these parasites are maintained as meiotically recombining populations in the laboratory, and individual parasites from each population may differ in genotype.

Cryopreservation of many viral, bacterial or protozoan pathogens simplifies research on these pathogens. However, while cryopreservation has been reported for schistosomes [1], it is inconsistent and cannot be used reliably for maintaining schistosome populations. Schistosome populations are therefore typically maintained

in the laboratory by continuous passage through the aquatic snail intermediate host, where asexual proliferation of larval stages occurs, and the rodent definitive host, where adult males and females pair, meiosis and recombination occur and eggs are produced.

Schistosome populations have been maintained in the laboratory for up to 80 years [2]. For example, the SmNMRI parasite population maintained by the Biomedical Research Institute (BRI) [3] was originally isolated from Puerto Rico in the 1940s [2]. Our laboratory maintains four different parasite populations: SmEG from Egypt, collected at an undetermined date (possibly in the 1980s) by US researchers and then established at the Theodor Bilharz Research Institute in Cairo in 1990 [4,5]. SmLE isolated in Brazil in 1965 [2], while SmBRE was acquired from Brazil in 1975 [6], and SmOR, a descendant from SmHR, which was isolated in Puerto Rico in 1971 [7]. Assuming five complete parasite generations per year (sexual and asexual), these parasite populations have been maintained continuously for ~400 (SmNMRI), ~160 (SmEG), ~285 (SmLE), ~235 (SmBRE), and 270 (SmOR) generations.

The genomic consequences of long-term laboratory passage in schistosomes are not known, but several authors investigated this question in the pre-genomic era. Fletcher et al. [8] examined enzyme polymorphism at 18 loci in individual worms. They measured mean heterozygosity per locus and observed that genetic variation within laboratory populations maintained from 1-40 generations was approximately half that observed in fresh parasite isolates. Minchella et al. [9] quantified genetic variation in a maternally inherited DNA element (pSM750) using restriction fragment length polymorphism (RFLP) of individual parasites from 14 laboratory isolates. They noted that parasites from the same laboratory isolate generally showed low variability. However, SmNMRI parasites exhibited extensive variation. Pinto et al. [10] found no variation between worms from a laboratory isolate (SmLE), but extensive variation within parasites derived from different Brazilian patients using random amplified polymorphic DNA (RAPD) analysis from three different primer sets. Hence, these studies reached rather different conclusions.

Efforts to sequence the genome of *S. mansoni* provided further insights. The *S. mansoni* genome was initially sequenced from pools of parasites from the SmNMRI population [11]. The genetic variation present within these populations contributed to an imperfect genome assembly: the resultant assembly was fragmented in > 19,000 scaffolds [11,12]. As a consequence, subsequent work used DNA isolated from worms of a single genotype, that were a product of snail infections with single miracidium larvae infections. This approach contributed to a much improved genome assembly, closing more than 40,000 gaps and assigning 81% of the data to chromosomes [13]. The current version 10 of the *S. mansoni* genome assembly was further improved using libraries constructed from schistosome infections derived from single miracidia, and encodes seven chromosomes, ZW sex chromosomes, and is ~391 Mb in size [14].

Phenotypic data provides further evidence that parasite populations may not be homogeneous. Davies et al. isolated parasites that shed low or high numbers of schistosome larvae (cercariae) from the snail host from the SmPR population [15]. Furthermore, they were able to select low and high shedding populations [16], indicating this phenotypic variation has a genetic basis. Similarly Le Clec'h et al. [17] demonstrated that the SmLE-PZQ-R population, which was selected for resistance to praziquantel (PZQ) in the SmLE population from Brazil, comprises a mixture of praziquantel (PZQ) resistant and sensitive parasites, as well as abundant variation across the genome [18].

The present study was designed to directly measure genomic variation within five laboratory schistosome populations. We speculated that either i) a low number of founders or inbreeding due to repeated laboratory passage could result in bottlenecks and therefore a loss of genetic variation or ii) sexual outbreeding could be sufficient to retain high levels of genetic variation. We generated 117 independent genome sequences from four schistosome populations maintained in our laboratory and from the widely used SmNMRI population maintained at the BRI. We compared variation in these laboratory populations with published exome sequence data from field collected *S. mansoni* parasites from Brazil, Niger, Senegal, and Tanzania [19]. We observed abundant genetic variation within laboratory populations, albeit less than half the variation observed in field collected parasites. However, laboratory and field collected parasites showed dramatic differences in pattern of variation, including the allele frequency spectrum, linkage disequilibrium, and effective population size ($N_e$). We evaluate the implications of these results for schistosome research.

## Results

### Summary of sequence data

We sequenced the genomes of 117 independent *S. mansoni* genotypes from five populations. This was done by sequencing cercariae larvae shed from each of 19–24 snails infected with single miracidia (for SmNMRI, SmOR, SmEG and SmLE) and by sequencing individual adult worms (for SmBRE). We retained 103 of 117 generated genome sequences from laboratory samples after quality filtering: 18 SmNMRI, 20 SmOR, 18 SmBRE, 24 for SmEG, and 23 for SmLE. The mean read depths for these samples was 32.8x (range: 10.0 – 143.8x), and we discovered 0.897 – 1.22 million single nucleotide polymorphisms (SNPs) in the laboratory populations. Detailed information about these variants is listed in Table 1.

In addition, we utilized 124 previously generated exome sequences from field collected parasites from Brazil ($n = 46$), Niger ($n = 9$), Senegal ($n = 23$), and Tanzania ($n = 45$) [19]. To make field and laboratory samples directly comparable, we filtered genotyped laboratory and field samples jointly, keeping only variants that were genotyped in ≥ 80% samples in each of the lab and field populations. This resulted in 131,207 autosomal variants (Table 2). Coverage statistics for each sample are listed in S1 Table.

### Principal component analysis (PCA) and admixture

We generated a PCA plot using 1.24 million MAF filtered, autosomal variants (MAF > 0.05) from our laboratory genome sequences (Fig 1A). This analysis identified five distinct clusters. While SmOR, SmEG, SmNMRI, and SmLE were separated along principal component 2, they clustered along principal component 1 and were distant from SmBRE. We used ADMIXTURE and plotted five populations, as $k = 5$ resulted in the smallest cross-validation score (Fig 1B). This analysis confirmed the presence of five schistosome populations with distinct allelic components.

We also generated a PCA plot containing both field and laboratory populations using 131,207 exon variants only (S1 Fig). Principal component 1 explains 7.84% of the variation and separates the Tanzanian field population from the other 8 populations. Principal component 2 explains 2.05% of the variation with SmBRE on one extreme and a cluster containing two west African field populations (Niger and Senegal) on the other. All remaining lab (SmEG, SmLE, SmOR, SmNMRI) and field (Brazil) populations form a closely related intermediate cluster.

**Table 1. Summary statistics of laboratory populations.**

| Population | No. Parasite genotypes | Mean coverage (Range coverage) | Total variants | SNVs | INDELS[1] | Autosomal SNPs | Mitochondrial SNPs | SNPs MAF > 0.05 |
|---|---|---|---|---|---|---|---|---|
| BRE | 20 | 71.1 (47.8, 143.8) | 8.97E+05 | 8.11E+05 | 8.55E+04 | 7.37E+05 | 7 | 1.26E+05 |
| EG | 24 | 24.8 (17.3, 38.3) | 1.22E+06 | 1.11E+06 | 1.10E+05 | 1.03E+06 | 9 | 8.69E+05 |
| LE | 24 | 23.4 (10.5, 44.5) | 1.01E+06 | 9.15E+05 | 9.65E+04 | 8.62E+05 | 7 | 5.23E+05 |
| NMRI | 19 | 26.3 (15.9, 38.5) | 1.08E+06 | 9.83E+05 | 9.35E+04 | 9.36E+05 | 2 | 7.23E+05 |
| OR | 21 | 24.4 (10.0, 42.2) | 1.07E+06 | 9.55E+05 | 1.19E+05 | 9.23E+05 | 5 | 6.40E+05 |

| Population | Number of samples | Synonymous coding | Non-synonymous coding | Intron | | Intergenic | |
|---|---|---|---|---|---|---|---|
| BRE | 20 | 9.65E+03 | 1.33E+04 | 4.64E+05 | | 4.27E+05 | |
| EG | 24 | 1.30E+04 | 1.53E+04 | 6.19E+05 | | 5.95E+05 | |
| LE | 24 | 1.03E+04 | 1.26E+04 | 5.12E+05 | | 4.96E+05 | |
| NMRI | 19 | 1.08E+04 | 1.33E+04 | 5.57E+05 | | 5.20E+05 | |
| OR | 21 | 1.10E+04 | 1.33E+04 | 5.51E+05 | | 5.20E+05 | |

[1]Mean INDEL size = -98, range (-369, 406).

**Table 2. Summary of variants used for comparison of laboratory and field collected schistosomes.**

| Population | Number of samples | Autosomal SNPs in CDS region | MAF filtered (> 0.05) |
|---|---|---|---|
| BRE | 18 | 6,790 | 399 |
| EG | 24 | 9,186 | 7,448 |
| LE | 23 | 7,203 | 4,040 |
| NMRI | 18 | 7,788 | 6,007 |
| OR | 20 | 7,669 | 5,283 |
| Brazil | 46 | 22,719 | 11,499 |
| Niger[1] | 9 | 17,848 | 16,784 |
| Senegal | 23 | 27,820 | 7,260 |
| Tanzania | 45 | 71,865 | 22,433 |

[1]MAF filtering removes variants present in just one copy ("singletons") in all populations except Niger. Presence of singletons may result in overestimation of variation in Niger.

## Nucleotide diversity in *S. mansoni* laboratory and field populations

The distribution of SNP variation across the genome in the laboratory populations is shown in Fig 2A. We calculated nucleotide diversity (π) in 25 kb windows (Fig 2B). Mean SNP numbers were similar across the chromosomes, but we observed an increase in the variance in SNP numbers at the chromosome ends (S2 Fig). This analysis revealed minimal diversity in the SmBRE population. While SmBRE had 1.26E + 05 segregating SNPs (MAF > 0.05), equivalent numbers for the other lab populations were 8.69E + 05 (SmEG), 5.23E + 05 (SmLE), 6.40E + 05 (SmOR) and 7.23E + 05 (SmNMRI) (Table 1).

We compared diversity in laboratory and field samples using a filtered dataset (Fig 2B). Laboratory populations (π = 1.52e-04 – 7.15e-04) showed 62% lower diversity than field populations (π = 9.06e-03 – 2.24e-03) (W = 360188499, $p < 0.001$). As previously documented, samples from Tanzania had the highest nucleotide diversity of all populations [19]. When we removed the Tanzania field population, genetic diversity in laboratory populations was 50% that found in field populations. The laboratory populations originating in Brazil (SmBRE and SmLE), showed 85% and 54% less variation than the Brazilian field population (Brazil).

We calculated the number of fixed derived variants in each population (Fig 2C) by identifying sites at fixation in *S. mansoni* populations, but absent from the outgroup (*S. rodhaini*). We observed more fixed derived SNPs in laboratory populations than field populations (Wilcoxon test, W = 2, p = 0.063). SmBRE, which showed the lowest diversity, showed the most fixed SNPs, while SmEG, which showed the highest diversity among the laboratory populations, carried the fewest.

We also plotted the average number of nucleotide differences per site ($D_{XY}$) for all pairwise combinations of populations to identify population specific regions of high or low diversity (S3 Fig). This includes 20 comparisons between lab and field populations, 6 comparisons between field populations and 10 comparisons between laboratory populations. These plots reveal genome regions on chr 2, 3, 4 and 6 with elevated $D_{XY}$ values. The top 50 pairwise comparisons identify five 25kb windows and contain 6 genes and include a mini-exon gene (Smp_159830 (MEG 2.3 isoform 1)) (S3 Fig).

## Tajima's D and allele frequency distributions

While four of five laboratory populations exhibited a positive Tajima's D, all field populations showed negative values (Fig 3A; Wilcoxon test; W = 17, *p* = 0.111). The exception to this was SmBRE, which had a negative Tajima's D like the field populations. We inspected allele frequency spectra in each population, to better understand why Tajima's D differs between populations. This revealed SNPs at intermediate frequencies were common in SmEG, SmLE, SmOR, and SmNMRI, whereas field populations (and SmBRE) had a high frequency of rare alleles (S4 Fig). We

PLOS Pathogens

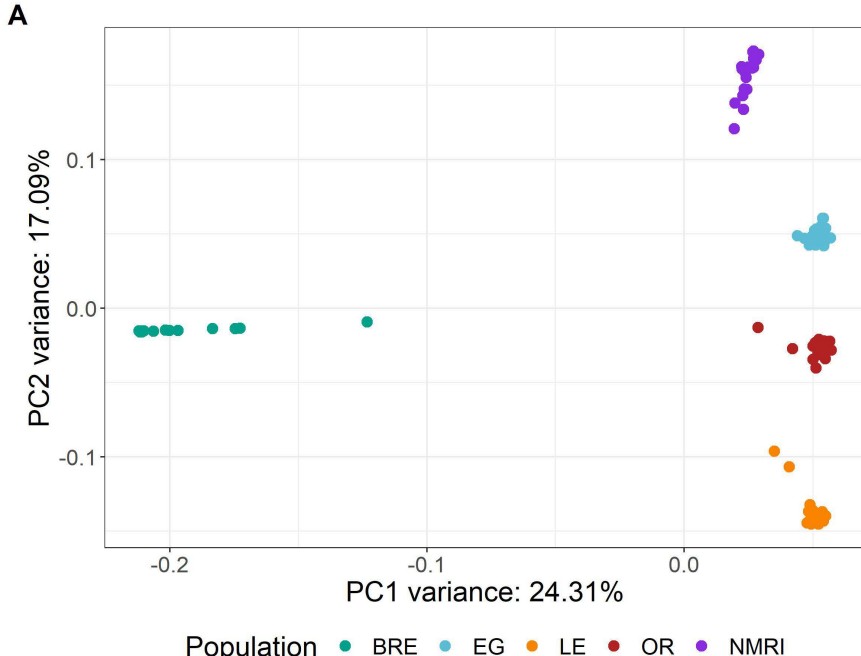

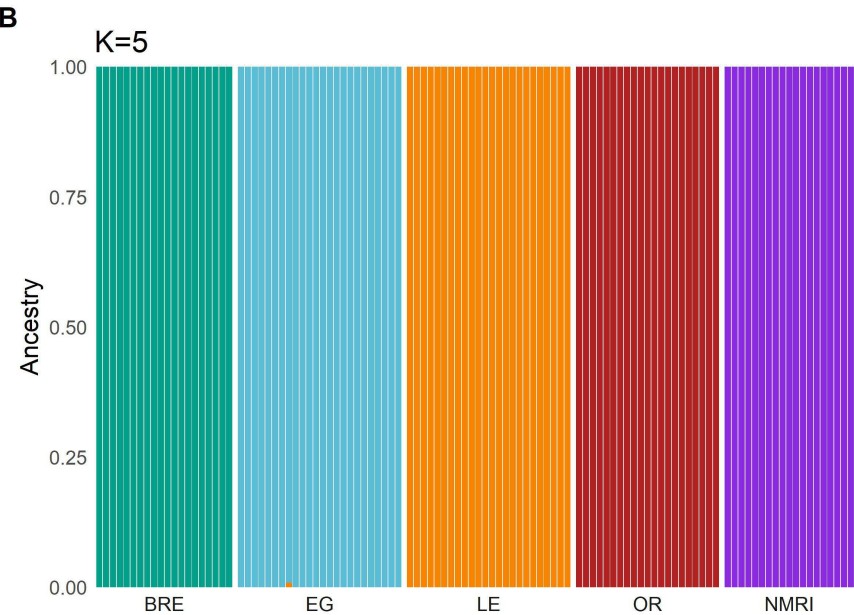

**Fig 1. Population structure in *S. mansoni* laboratory populations. (A)** PCA plot showing clustering of sequenced *S. mansoni* laboratory populations. **(B)** Admixture analysis with *k* = 5 populations.

plotted the empirical cumulative distribution (ECDF) of allele frequencies for each population (Fig 3B) and measured Kolmogorov-Smirnov statistics for all using pairwise comparisons. Observed mean statistics for comparisons of field and lab populations, were significantly higher than those observed in all but 12 of 10,000 permutations (one side p-value = 0.0012), demonstrating significant differences in allele frequency spectra of field and lab parasite populations.

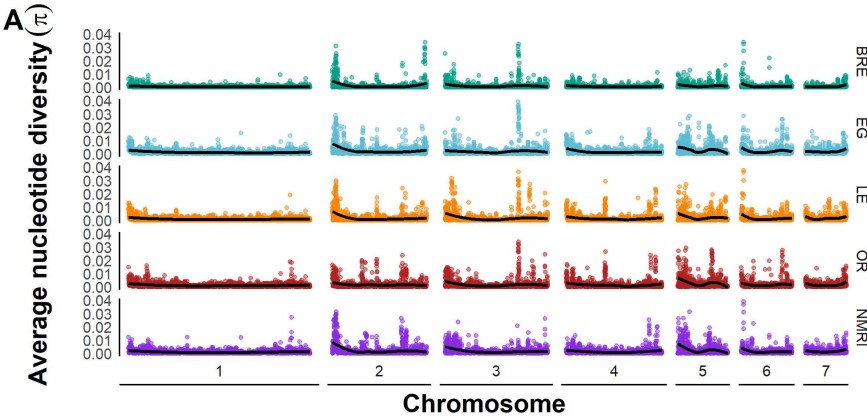

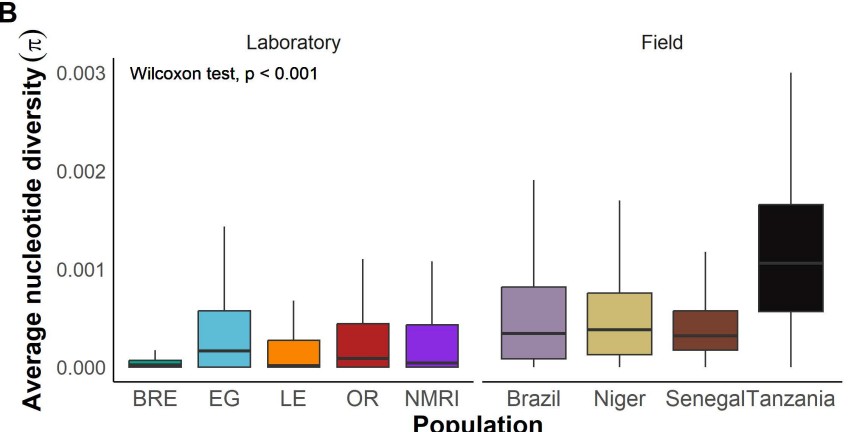

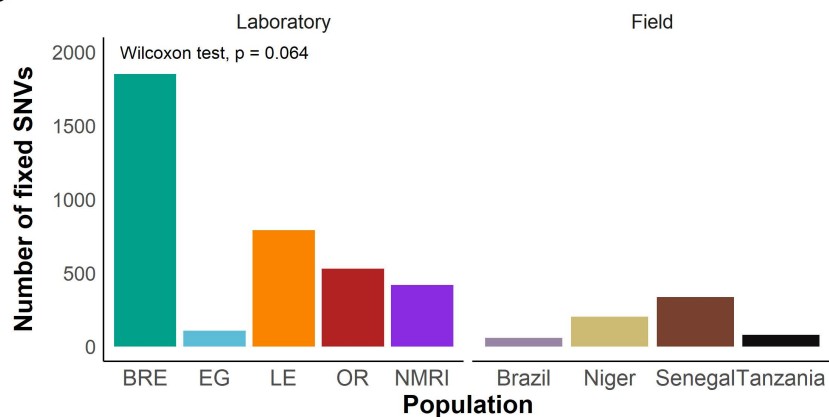

**Fig 2. Comparable nucleotide diversity in field and laboratory populations. (A)** Average nucleotide diversity (π) across the whole genome for each laboratory population calculated in 25 kb windows and plotted for each autosome. The line indicates a LOESS smoothed curve. **(B)** Box and whisker plot showing nucleotide diversity (π) in 25 kb windows across the CDS in laboratory and field populations. Outliers are not shown. Wilcoxon tests compares π from laboratory and field populations **(C)** Histogram showing numbers of derived fixed alleles in each of the laboratory and field populations. Wilcoxon tests compares numbers of derived fixed alleles from laboratory and field populations.

**A**

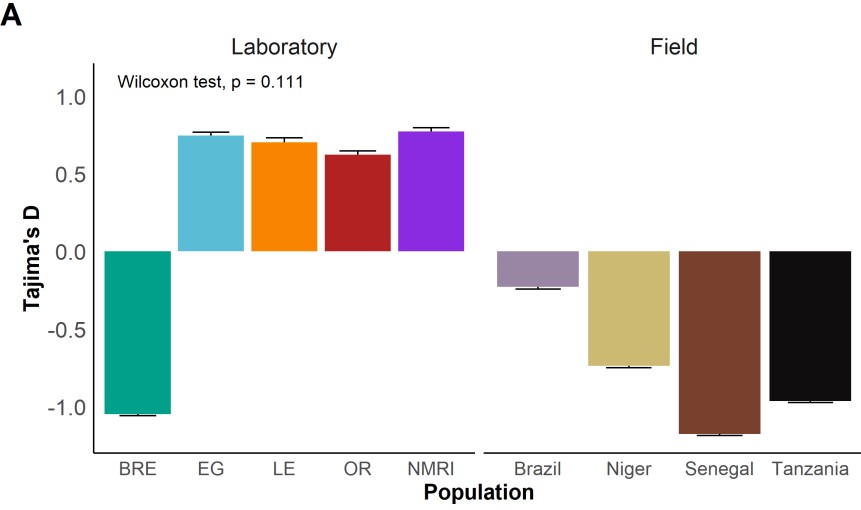

**B**

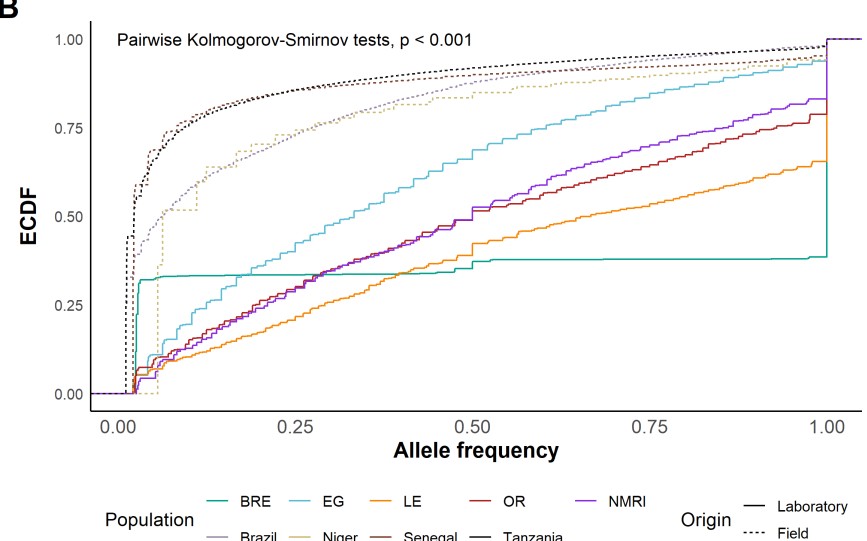

**Fig 3. Indicators of recent bottlenecks in laboratory populations. (A)** Bar plots showing mean and standard error of Tajima's D in each population. We used a Wilcoxon test to compare means of Tajima's D in field and laboratory populations. **(B)** Line plot showing the empirical cumulative distribution function (ECDF) of allele frequencies in each population. A permutation-based Kolmogorov-Smirnov test was used to compare field vs laboratory distributions (see S4 Fig).

## Linkage disequilibrium in laboratory and field populations

We calculated linkage disequilibrium (LD) for each *S. mansoni* population and estimated LD decay with physical distance between markers from pairwise $r^2$ values. As we only retained 399 common (MAF > 0.05) exonic SNPs in the SmBRE population, we used all autosomal variants to calculate LD decay in the laboratory populations. Fig 4A shows slower LD decay in four out of the five laboratory populations compared to the field populations. To compare LD decay curves, we measured the distance at which LD is reduced to $r^2 = 0.5$ ($LD_{0.5}$, Fig 4B) [20]. LD decayed extremely rapidly in the Tanzanian parasite population ($LD_{0.5} = 9$ bp). LD decayed uniformly in the Nigerien, Senegalese, and Brazilian populations, with $LD_{0.5}$ ranging from 1,000–9,543 bp. LD decay was slower in the laboratory populations (Wilcoxon test, W = 17, p = 0.111), with $LD_{0.5}$ ranging from 72 kb to 180 kb in SmEG, SmLE, SmNMRI, and SmOR. In stark contrast to other laboratory

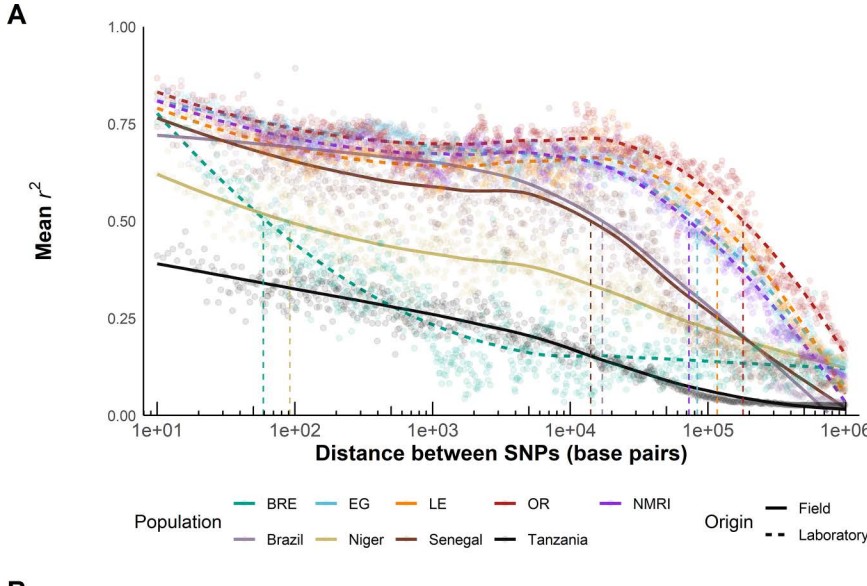

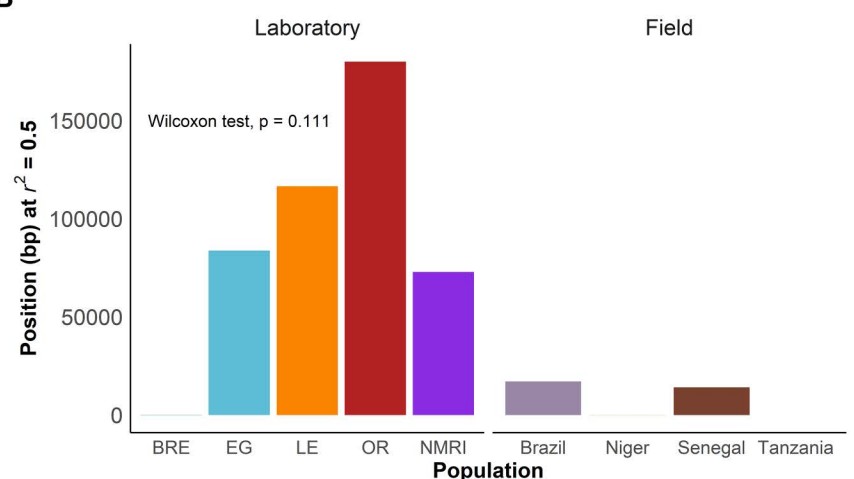

**Fig 4. Slower LD decay in laboratory populations. (A)** $r^2$ showing LD decay with physical distance between all autosomal SNPs in laboratory populations and exonic SNPs in field populations along the chromosomes. The points show mean values calculated over 1 kb windows while the fitted lines show relationship between distance and LD decay in each population. The plots are on a log scale. **(B)** Bar plot showing position when $r^2 = 0.5$ ($LD_{0.5}$) for field and laboratory populations. A Wilcoxon test was used to compare $LD_{0.5}$ in field and laboratory populations. S5 Fig shows LD calculated using exonic SNPs, while S6 Fig shows LD between unlinked markers on different chromosomes.

populations, SmBRE exhibited very rapid LD decay ($LD_{0.5} = 59$ bp). We also calculated LD using exonic SNPs only to ensure that the differences observed did not result from use of different marker sets in field and laboratory populations (S5 Fig). This confirmed slower LD decay in laboratory than field populations (Wilcoxon, W = 17, $p = 0.111$), with the exception of SmBRE.

## Population size

We used our sequencing data to predict the current effective population size ($N_e$) based on either linkage disequilibrium (NeEstimator) or sibship frequency (COLONY). NeEstimator computed effective population sizes ranging from 2 – 258

in the laboratory and 3,174 (Brazil) – infinity (Niger, Senegal, Tanzania) in the field populations (Fig 5A), while COLONY reported $N_e$ values from 5 – 123 for laboratory populations and 3,612 – infinity for field populations (Fig 5B). Both NeEstimator and COLONY identified SmNMRI and SmLE as having the highest $N_e$ estimates among the laboratory populations, while SmBRE had the lowest $N_e$. $N_e$ estimates for laboratory populations using both approaches were correlated ($R^2 = 0.96$, $p = 0.020$). $N_e$ estimates were at least 12-fold greater in field than in laboratory schistosome populations with NeEstimator and at least 29-fold greater with COLONY.

Using our life cycle maintenance records, we estimated the census size ($N_c$) of our four laboratory schistosome populations over time and calculated the harmonic mean of each population [21]. This was done by estimating the number of parasite genotypes used to infect hamsters for each laboratory maintenance cycle over a seven-year period (S7 Fig). We

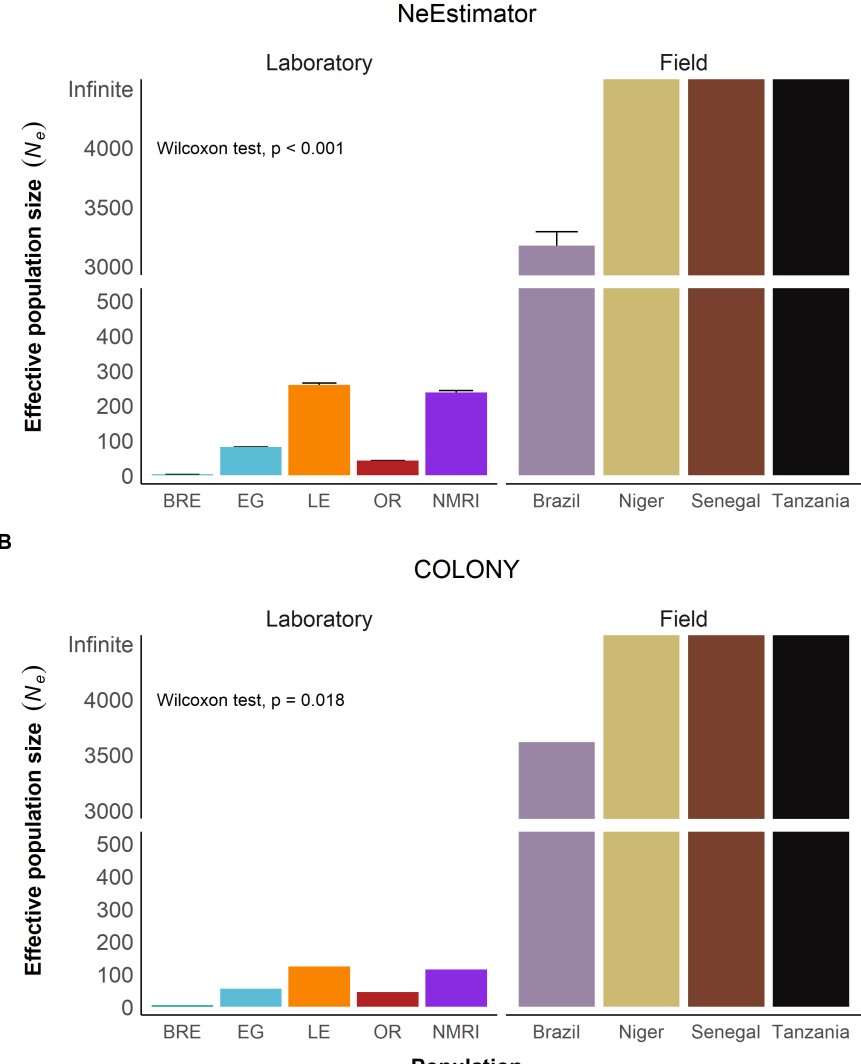

**Fig 5. Reduced effective population size in laboratory populations.** Bar plots showing effective population size $N_e$ calculated with **(A)** NeEstimator and **(B)** COLONY. The y-axis is split to show both high and low $N_e$ values clearly. The error bars represent a 95% confidence interval. We used Wilcoxon tests to compare $N_e$ in laboratory and field populations. Infinite values were set to 100,000 for this purpose.

did not have census data for the SmNMRI population maintained at BRI. Census size remained relatively consistent in SmLE, SmOR, and SmEG. However, population size increased in SmBRE parasites starting in 2021 (S7A Fig), as a result of a laboratory contamination event [22]. SmLE had the highest census with 157 genotypes, followed by SmOR (137) and SmEG (132), and SmBRE (93) (S7 Fig). Population size data is summarized in Table 3.

### Simulations of genomic diversity in populations of different size

We conducted simulations to examine how population size impacts retention of diversity in schistosomes (Fig 6). When the number of parasite genotypes ($N$) = 5, where N is the number of adult worm genotypes in each generation, simulations show that autosomal diversity is reduced by >99% in 91 generations. When $N = 100$, we observe a 63.4% reduction relative to the progenitor population after 400 generations. When $N = 200$, the reduction 39.9% after 400 generations. The average effective population size ($N_e$) in our laboratory populations is 124, as calculated by NeEstimator, and 68, as calculated using Colony (Table 3).

## Discussion

### High levels of genetic diversity in most laboratory schistosome populations

We sequenced parasites from five different laboratory-maintained *S. mansoni* populations and compared them to four field populations from Africa and South America. Our genomic data revealed 0.897 – 1.22 million variants segregating within the five laboratory populations. This is equivalent to one variant every 321–436 bp. Furthermore, our study revealed 62% lower nucleotide diversity (π) in exome data from laboratory-maintained schistosome populations than from field populations. Despite repeated passage over 30 – 80 years (~150–400 generations, assuming five generations per year), high levels of genetic diversity remain in many laboratory schistosome populations.

Studies of free-living animals and plants have compared the genetic composition of different wild and domesticated/ farmed species. Domestication of animals or plants often involves a low number of founders and the selection for specific traits [23–27]. Nucleotide diversity (π) is reduced in domesticated species populations by between 33 and 98% relative to wild populations (S2 Table) [28,29]. Laboratory *S. mansoni* populations fall close to the center of this range, with π being 62% lower relative to wild populations, a level of diversity reduction comparable to Mediterranean brown trout [30] and

Table 3. $N_c$, $N_e$ estimates and ratios.

| Population | Census | Census CI95 (L) | Census CI95 (U) | NeEstimator | NeEstimator CI95 (L) | NeEstimator CI95 (U) | COLONY | COLONY CI95 (L) | COLONY CI95 (U) | NeEstimator *Ne/Nc* | COLONY *Ne/Nc* |
|---|---|---|---|---|---|---|---|---|---|---|---|
| BRE | 93 | 79 | 114 | 2 | 2 | 2 | 5 | 2 | 20 | 0.02 | 0.05 |
| EG | 132 | 111 | 161 | 81 | 80 | 81 | 55 | 33 | 112 | 0.61 | 0.42 |
| LE | 157 | 127 | 205 | 258 | 253 | 264 | 123 | 71 | 417 | 1.65 | 0.79 |
| OR | 137 | 112 | 174 | 42 | 42 | 42 | 45 | 27 | 99 | 0.31 | 0.33 |
| NMRI | | | | 237 | 232 | 242 | 114 | 60 | 581 | | |
| Brazil | | | | 3,174 | 3,064 | 3,291 | 3,612 | 1,211 | Infinite | | |
| Niger | | | | Infinite | Infinite | Infinite | Infinite | 1 | Infinite | | |
| Senegal | | | | Infinite | Infinite | Infinite | Infinite | 1 | Infinite | | |
| Tanzania | | | | Infinite | Infinite | Infinite | Infinite | 1 | Infinite | | |

Measures of Census population size ($N_c$), and are based on the estimated number of parasite genotypes present in infected snails used to infect hamsters. $N_c$ measures are measured from colony maintenance data collected over the past 7 years only, and are no available for field collected parasites. Effective population size estimates (Ne) are estimated from sequence data using two different estimator approaches (NeEstimator and Colony).

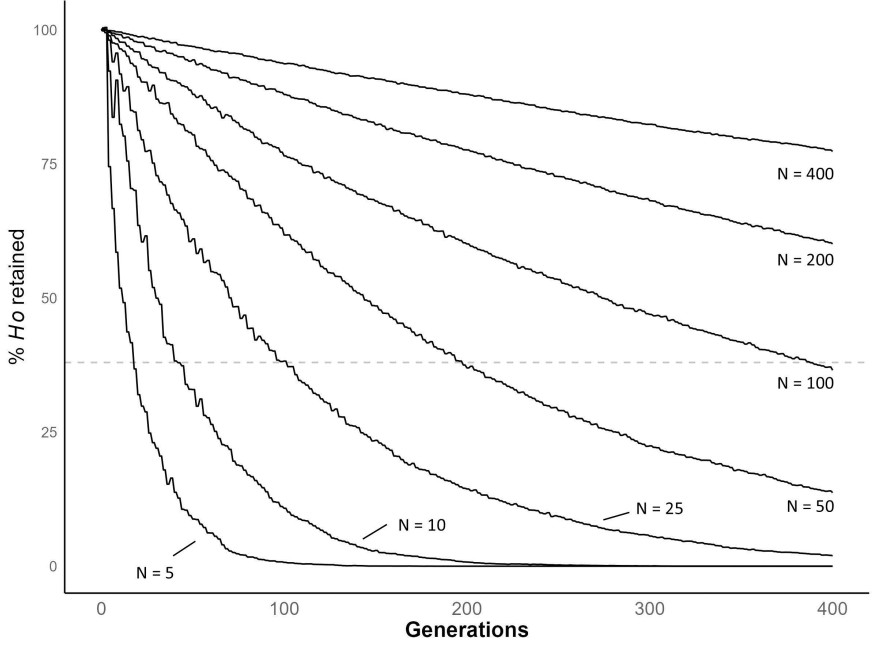

**Fig 6. Bottleneck simulation over 400 generations.** Line plot showing simulated reduction in genetic diversity of schistosome populations of different sizes over 400 generations. We used constant N ranging from 5 – 400. $H_0$ indicates heterozygosity at time 0. The horizontal dashed line (38%) shows of the % reduction in diversity observed in our laboratory populations when compared to field populations.

sunflower [31]. The relatively high levels of variation observed in laboratory schistosome populations may result from: (i) the relatively large size of *S. mansoni* founder populations; laboratory schistosome populations are typically founded by collecting eggs from one or more patients, each of which may be infected with hundreds of adult worms [32]; and (ii) that laboratory schistosome populations are maintained in large populations to prevent loss during laboratory maintenance. Our census estimates show that numbers of independent schistosome genotypes used to infect hamsters ranges from 93-157 (harmonic means) in our laboratory populations.

The level of variation retained within populations is dependent on the size and duration of population bottlenecks as demonstrated with our population bottleneck simulation [33]. Our $N_e$ estimates are 2 – 258 with NeEstimator and 5 – 123 with COLONY, while our census ($N_c$) estimates range from 93 to 157. That laboratory populations show 62% less nucleotide diversity ($\pi$) compared to field populations is generally compatible with the simulation results when N ≈ 100. This is broadly consistent with the observed $N_c$ or $N_e$ values estimated. However, we note that factors other than demographics may maintain genetic variation: both the action of balancing selection [34] or preferential mating between unrelated parasites [35] may also act to retain genetic variation in laboratory parasite populations. Nevertheless, given sufficient breeding adults (~100) during laboratory passage, maintenance of genetic variation within these schistosome populations over multiple years to laboratory maintenance is perhaps not surprising.

While our study indicated lower levels in nucleotide diversity in laboratory compared with field schistosome populations, there were dramatic differences in the *pattern* of variation in laboratory and field populations. The patterns observed are consistent with strong bottlenecks during establishment and maintenance *of S. mansoni* colonies. We observed (i) lower numbers of rare alleles in laboratory populations: this is reflected in the positive Tajima's D for four of the five laboratory populations, while the field populations show negative Tajima's D. This is furthermore confirmed by the allele frequency spectra, which show a comparative deficit in rare alleles and more alleles at intermediate frequencies compared to field populations. Population contraction in the laboratory is the most likely cause of the allele frequency spectra observed

as intermediate frequency alleles are more likely to survive bottlenecks [36]. (ii) more fixed derived SNPs in laboratory populations: this is consistent with increased drift in small populations. Notably, we observed the greatest number of fixed SNPs in SmBRE, which showed the lowest diversity and Ne estimates, and the lowest number of fixed SNPs in SmEG, which showed the highest diversity and Ne estimates (ii) Differences in LD decay: with the notable exception of SmBRE, we observed 6 – 19-fold slower decay of LD with physical distance on chromosomes in laboratory populations compared with field populations. This reduction is expected given the increased levels of sib-mating, genetic drift, and reduced total number of meiotic recombination events in small laboratory populations.

### The exception: SmBRE is depauperate and shows low fitness

We have studied the SmBRE laboratory population extensively. These parasites typically show reduced snail infectivity, lower cercarial shedding and virulence in the intermediate snail host, and reduced immunopathology in the mouse [37–40]. One possible explanation for low fitness of SmBRE is inbreeding depression. In line with this, we found that nucleotide diversity ($\pi$) was two- to threefold lower in SmBRE than in the other four laboratory populations; SmBRE also had the lowest estimates for effective population size ($N_e$), and elevated levels of fixed SNPs (Table 1). We sequenced adult worms for SmBRE rather than cercariae. We were concerned that some SmBRE adults sequenced might be derived from clonal cercariae emerging from a single snail, which would downwardly bias our diversity measures. However, none of the SmBRE parasites showed elevated relatedness (r > 0.9) so this explanation can be eliminated.

However, other population genetic results for SmBRE were entirely unexpected. While SmEG, SmNMRI, SmLE, and SmOR showed strongly positive Tajima's D, SmBRE had a strongly negative Tajima's D and site frequency spectra like the field collected populations. We do not know what features of SmBRE demography might have contributed to this.

LD analysis for SmBRE produced the most puzzling result and showed a pattern that was radically different from the other laboratory populations. Given the low genetic diversity and effective population size ($N_e$) in SmBRE, we had expected to see the slowest rate of LD decay in this population. However, LD decayed very rapidly in SmBRE, and was higher than three of the four field populations examined. A genetic map for *S. mansoni* revealed that 1cM = 227 kb (95% CI 181–309kb) [41]. A possible explanation for the rapid decay of LD is that the recombination rate may be higher in SmBRE than in other laboratory populations. Analysis of further *S. mansoni* genetic crosses involving SmBRE could be used to explore this hypothesis. It is known that recombination rates can vary both across the genome and among populations of the several species [42,43].

### Implication for schistosome research

That four of five laboratory-maintained schistosome populations retain abundant genetic variation has several important implications for schistosome research:

**Phenotype measures in individual worms.** Current research on schistosome parasites, including developmental, immunological, transcriptomic, or drug response studies, utilizes pools of genetically variable worms rather than homogeneous inbred parasite lines [44–47]. As a consequence, these studies capture average population phenotypes and underestimate variation in the traits studied. For example, our laboratory has recently documented a significant impact of parasite population on immunopathological parameters, including spleen/liver weight and fibrosis [37]. However, we recognize that these impacts are likely to be underestimated, as our studies, like many others, utilize genetically variable laboratory populations. Analysis of praziquantel response provides a dramatic example. The SmLE-PZQ-R laboratory population, selected for praziquantel resistance, shows 14-fold increase in drug resistance relative to SmLE population from which it was selected. However, SmLE-PZQ-R is a mixture of both PZQ sensitive (PQZ-S) and PZQ resistant (PQZ-R) parasites that differ by > 377 fold in drug response [18]. We suspect other parasite phenotypes may show equally dramatic variation when measured in individual worms rather than diverse populations. Open source tools like the Single Worm Analysis of Movement Pipeline (SWAMP) [48] and wrmXpress [49] now offer the capability to

accurately measure drug response phenotypes in individual worms, while transcriptomic variation can be measured in single worms or single cells [50–53]. We encourage researchers to shift their focus from genetically diverse populations to individual parasites for clearer measurement of parasite phenotypes.

**Genome-wide association studies (GWAS).** Schistosome parasites show abundant phenotypic variation in a wide range of traits [54]. These include cercarial shedding [16,38,39,55–57], host specificity [58–60], and drug resistance [18,61–64]. Our laboratory is specifically interested in understanding the genetic basis of phenotypic traits in schistosomes, and we have primarily used genetic crosses and linkage analysis for this purpose [54]. That high levels of genetic variation are found within laboratory populations allows us to use a second powerful mapping approach (GWAS) to identify genes underlying specific traits. GWAS is considerably simpler than linkage analysis, because conducting two-generation (F2) genetic crosses is not required. Furthermore, GWAS more effectively examines variation across multiple individuals within populations, while genetic crosses examine differences between the two parents only, so samples genetic and phenotypic variation less effectively. Le Clec'h et al.'s [18] work on PZQ resistance provides strong proof-of-principal for use of GWAS approaches for schistosomes using laboratory populations. Their GWAS study used single worm measures of drug response in the SmLE-PZQ-R population and then sequenced pools of PZQ-S and PZQ-R worms showing extreme drug response phenotypes to determine the genome regions involved [18,65].

GWAS relies on association (LD) between trait loci and surrounding genetic markers. We observed much slower decay in LD in four out of five laboratory-maintained schistosome populations than observed in the field. GWAS studies in laboratory populations are therefore likely to generate much broader peaks [66,67]. For example, in the GWAS of praziquantel resistance locus, the genome region mapped spanned 5.72 mb and 137 genes [65]. Broad peaks have some advantages, as such peaks are unlikely to be missed if they are situated in genome regions that are difficult to genotype. However, broad peaks containing multiple genes make the task of identifying the causative locus much harder. We note that the extremely rapid decay in LD observed in some field populations (e.g., Tanzania) suggests that GWAS using freshly isolated parasite populations collected from infected patients may result in narrow peaks and allow identification of candidate regions with greater precision.

**Reproducibility at different institutions.** Several laboratories maintain the same and/or different schistosome populations as examined here. The literature often refers to these schistosome populations as strains or lines, akin to bacterial clones or inbred mice, and so the assumption is that they will produce similar results at different institutions. However, bottlenecks and low $N_e$ will result in genetic drift, and divergence between populations at different institutions. Such changes are likely to affect reproducibility, as is the case with non-model rodents [68]. Accessing schistosome parasites through the BRI [3] increases short-term consistency and reliability, but even parasites obtained from BRI in different years may vary due to genetic drift. While genetically variable laboratory populations have advantages for some genetic analyses (e.g., see "Genome-wide association studies"), one possible solution to increase repeatability in laboratory experiments might be to establish inbred parasite lines by serial inbreeding over a minimum of seven generations to reach 99% homozygosity. Such inbred lines have been used for snails [69] and mice [70]: the addition of inbred schistosome lines would allow precise dissection of parasite host interactions across the parasite life cycle. However, our experience with SmBRE illustrates that highly inbred populations may suffer from inbreeding depression and reduced fitness, posing a significant challenge to overcome.

## Relevance to other helminths

Recent work suggests that the degree of genetic variation in other laboratory-maintained helminths could be underestimated as well. Stevens et al. [34] showed that *Heligosomoides bakeri,* a commonly used model nematode of rodents, retains extensive genetic diversity despite laboratory maintenance for 70 years [34]. Even in the selfing nematode *C. elegans,* long-term balancing selection maintains genetic variation to increase fitness and survival [71]. Studies of genetic variation filarial nematodes are particularly informative [72] (Table 4). Populations of the mosquito transmitted filarial

**Table 4. Nucleotide diversity in *Brugia malayi* and *B. pahangi*.**

| Parasite population | Origin | N | Sequence coverage | No. SNV | π | ± 1 SD | Accession |
|---|---|---|---|---|---|---|---|
| **Brugia malayi** | | | | | | | |
| CDRI, Lucknow, India | Lab[a] | 4 | 95.94 | 108,035 | 0.0012 | 0.0015 | SRR3111731, SRR3111738, SRR3111864, SRR3112012 |
| TRS Jird1 | Lab[a] | 4 | 95.86 | 87,661 | 0.0013 | 0.0015 | SRR111504, SRR3111510, SRR3111514, SRR3111517 |
| FR3 Jird Bmalayi | Lab[a] | 4 | 95.26 | 88,357 | 0.0014 | 0.0015 | SRR3111544, SRR3111568, SRR3111579, SRR3111581 |
| Liverpool Jird | Lab[a] | 4 | 92.23 | 88,920 | 0.0013 | 0.0015 | SRR3111629, SRR3111630, SRR3111634, SRR3111636, SRR3111640 |
| WashU Jird | Lab[a] | 6 | 94.86 | 93,937 | 0.0013 | 0.0012 | SRR3111318, SRR3111319, SRR5190289, SRR5190290, SRR3111488, SRR3111493, SRR3111498, SRR5190287, SRR5190288 |
| Thai Jird | Lab[b] | 4 | 92.08 | 190,455 | 0.0013 | 0.0015 | SRR12884294, SRR12884293, SRR12884292, SRR12884291 |
| **Brugia pahangi** | | | | | | | |
| FR3, University of Georgia, Athens, Georgia, USA via BEI | Lab[c] | 3 | 87.07 | 219,183 | 0.0025 | 0.0031 | SRR13482041, SRR13482040, SRR13482039 |
| University of Malaya, Kuala Lumpur, Malaysia | Field[d] | 3 | 90.97 | 507,464 | 0.0042 | 0.0044 | SRR7226912, SRR7227476, SRR7227477, SRR7227478, SRR7227479 |
| FR3, University of Wisconsin Oshkosh, Oshkosh, Wisconsin, USA | Lab[c] | 4 | 94.56 | 204,101 | 0.0023 | 0.0030 | SRR12884296, SRR12884297, SRR12884298, SRR12884299 |

These data are reanalyzed from Mattick et al. [72]. Note that SNP calls in Mattick et al. [72] and this paper were made with different software tools (GATK for Mattick et al [72] and bcftools v1.22 in this paper). Hence the number of SNPs called may not match precisely.

[a]Originally established from an infected human from Malaysia in the 1960s, and distributed to several laboratories [73,74].

[b]Originally established from an infected human from Thailand in the 1980s [75].

[c]Originally established from a green leaf monkey in the 1970s [76].

[d]Adult worms isolated from a naturally infected cat [77].

nematodes *Brugia malayi* established from microfilariae from a single infected human have been maintained in several laboratories since the 1960s, while several populations of *B. pahangi* (originally collected from a green leaf monkey) have been maintained since the 1970s. Yet these laboratory filarial parasites remain genetically diverse: laboratory *B. malayi* individuals carried on average 108035–190445 segregating SNPs, while laboratory *B. pahangi* colonies contained 204,101–219,183 SNPs. *B. pahangi* adults retrieved from cats provide a comparison from natural infections: these carried segregating 507,464 SNPs. Hence, *B. pahangi* maintained for ~50 years in the laboratory still carry 50–60% of the genetic variation found in field collected worms (Table 4), similar to our results from *S. mansoni*. We expect that sequencing of other laboratory-maintained helminths, such as *Trichuris muris* or *Strongyloides* spp. may also reveal substantial genetic diversity, providing new research opportunities for a wide range of model helminth parasites.

## Limitations of this study

Paired field and laboratory populations from the same location are most informative for examining the impact of laboratory culture or domestication. These were not available here, so we compared laboratory sequence variation with that from previously published, but independent field collected samples. We used Illumina short read sequencing for this work. Highly variable genes are difficult to align to a reference sequence, so are poorly genotyped in Illumina-based resequencing studies due to poor read

mapping in these regions. It is therefore likely that we significantly underestimated diversity in this study. For a more exhaustive evaluation of genetic variation within laboratory schistosome populations, long read (Nanopore or PacBio) sequencing, Hi-C and *de novo* assembly will be needed [34]. For the same reason, our study was not well powered to detect islands of genetic variation, suggestive of balancing selection, as observed in *C. elegans* [71] and *H. bakeri* [34] (see "Relevance to other helminths").

An additional potential concern is the comparison of whole genome sequence from laboratory samples with exome capture for field samples. Exome capture methods may poorly sequence divergent alleles resulting in underestimation of variation. In reality, this potential bias is expected to be minimal, because capture methods typically allow efficient sequencing of alleles ≤10% divergent from the RNA baits used [78].

## Materials and methods

### Ethics statement

We utilize Syrian hamsters as the rodent host for maintaining schistosome parasites. This study was performed in accordance with the Guide for the Care and Use of Laboratory Animals of the National Institutes of Health. The protocol was approved by the Institutional Animal Care and Use Committee of Texas Biomedical Research Institute (permit number: 1419-MA).

### Recovery of *Schistosoma mansoni* miracidia and snail infections for sample generation

For each parasite population (except SmBRE), we collected pools of cercariae shed from snails infected with single miracidia larvae. This ensures that each parasite sequenced is an independent genotype. We extracted gDNA from cercarial larvae in lieu of adult worms for two reasons: (i) adult schistosome females carry fertilized eggs which would result in mixed genotype sequences, and (ii) we wanted to avoid the sampling of identical adult worms derived from clonal cercariae from a single snail. In brief, we recovered *S. mansoni* eggs from livers of infected Golden Syrian hamsters as previously described [79] and infected *Biomphalaria glabrata* (line Bg36 for SmOR and SmLE) and *B. alexandrina* (for SmEG) snails by placing individuals in 24-well plates with a single miracidium. Plates were placed under a light source overnight before putting the snails in trays covered with a clear plastic lid. The lids were exchanged for a dark lid three weeks post infection to prevent cercarial shedding.

### Sample generation for SmBRE

Preliminary analyses for this project revealed that our SmBRE population was contaminated with SmLE [80]. We therefore extracted gDNA from adult schistosome parasites previously collected during life cycle maintenance, prior to contamination. Individual male worms were processed as described below. To avoid obtaining mixed genotype eggs, we decapitated individual female worms and extracted gDNA for sequencing with Chelex solution following an established protocol [41]. All samples were whole-genome amplified as described below.

### Collection of *S. mansoni* cercariae and gDNA extraction

We placed all snails into 24-well plates and shed them for 2 hours under light 28 days post infection. The content in each individual well was collected, transferred into microtubes, and spun down at $500 \times g$ for 5 minutes to pellet the cercariae. We removed supernatant before flash-freezing cercariae in liquid nitrogen. Samples were stored at -80°C until gDNA extraction with the DNeasy Blood & Tissue Kit (Qiagen, Germantown, MD, USA) according to manufacturer instructions (tissue lysis for 2 hours at 56°C). We quantified gDNA using a Qubit dsDNA BR Assay Kit (Invitrogen, Carlsbad, CA, USA). We used the GenomiPhi V2 DNA Amplification Kit for whole genome amplification (WGA) of samples with gDNA yield < 200 ng (Cytiva, Marlborough, Massachusetts, USA). We have previously demonstrated high accuracy (99.55%) of genotyping from WGA of schistosome material, by comparing genotypes of F1 larval parasites with their parents [81].

## gDNA Library preparation and sequencing

We used the KAPA HyperPlus Kit with library amplification (Roche, Indianapolis, IN, USA) to generate whole genome libraries with 200–400 ng of input material. We followed the manufacturer's instructions with the following modifications: we fragmented the samples for 25 minutes, amplified libraries using six PCR cycles, and we performed library size selection using a ratio of 0.6X (30 µl beads) for the first size cut and 0.8X (10 µl beads) for the second size cut. We assessed the library profile with TapeStation 4200 D1000 ScreenTape (Agilent, Santa Clara, CA, USA) (average library size: 455) and quantified all libraries with the KAPA Library Quantification Kit (Roche, Indianapolis, IN, USA) (average library concentration: 43 nM). Pooled samples were sent to Admera Health and sequenced on a NovaSeq S4 (one pool with 40 samples) or NovaSeq X Plus (3 pools with 18–19 samples) platform (Illumina) using 150 bp paired-end reads.

## Computational environment

We used conda version 23.1.0 to manage environments and download packages used in the analysis. Data was processed in R 4.2.0 using *tidyverse* v1.3.2, and plots were generated with *ggplot* v3.4.2. All shell and R scripts written for this project are available at https://github.com/kathrinsjutzeler/sm_single_gt and Zenodo https://doi.org/10.5281/zenodo.10672478.

## Genotyping

We used trim_galore v0.6.7 [82] (-q 28 --illumina --max_n 1 --clip_R1 9 --clip_R2 9) for adapter and quality trimming before mapping the sequences to version 9 of the *S. mansoni* reference genome (GenBank assembly accession GCA_000237925.5) with BWA v0.7.17-r118 [83] and the default parameters. We used GATK v4.3.0.0 [84] for further processing of the sequences. First, we removed all optical/PCR duplicates with MarkDuplicates. Next, we called single nucleotide polymorphisms (SNPs) with HaplotypeCaller and GenotypeGVCFs on a contig-by-contig basis, which we combined for each individual and finally merged into a single VCF file for all sequences, including the ones from previously processed field samples [19]. At this point, we lifted the file over to v10 of the *S. mansoni* reference genome (Wellcome Sanger Institute, project PRJEA36577) using LiftoverVcf. We used VariantFiltration with the recommended parameters (FS > 60.0, SOR > 3.0, MQ < 40.0, MQRankSum < -12.5, ReadPosRankSum < -8.0, QD < 2.0) and VCFtools v0.1.16 [85] for quality filtering. For variant statistics of genomic data from laboratory populations, we removed sites with quality < 15, read depths < 10, and missingness > 20% and individuals with a genotyping rate < 50%. For the combined laboratory/field population analyses of exome data, we used bedtools intersect (v2.31.0) [86] to keep sites overlapping with the exome probes used for field samples. We then filtered each population individually by removing i) sites with quality < 15, read depth < 10 and > 20% missingness, and ii) individuals with a genotyping rate < 50%. Finally, we filtered [86] the population files to keep high quality sites that were scored in 80% of individual parasites from each of the laboratory and field populations.

## Principal component analysis (PCA) and admixture

We used the snpgdsPCA() function from the *SNPRelate* v1.30.1 [87] R package to generate the PCA matrix and ADMIXTURE v1.3.0 [88] to estimate population ancestry for which we examined between $k = 1$ and $k = 10$ populations. In the end, we chose the model with the smallest cross validation score and used Q estimates as a proxy for ancestry fractions.

## Summary statistics, Tajima's D, and nucleotide diversity (π)

We calculated coverage statistics with samtools v1.9 [89] and mosdepth v0.3.6 [90]. We used VCFtools to calculate Tajima's D in windows of 25 kb using autosomal variants in each population separately. We generated a VCF file containing both variant and non-variant sites from the genotyped GATK database to calculate nucleotide diversity (π) in 25 kb windows with pixy [91]. To compared relative nucleotide diversity (π) in laboratory and field populations, we measured the mean π in lab populations

(by calculating π for each lab population, then taking the average) and comparing to the mean π in Field populations. We used the same VCF file to calculate the average number of nucleotide differences per site ($D_{XY}$) for all pairwise combinations of populations using pixy [91]. We calculated numbers of fixed derived SNPs in each population by comparison to *S. rodhaini* (ERR114786, ERR310938, ERR7978134, ERR7978135, ERR7978144, SRR16526444, SRR16526443, SRR16526442, SRR16526441, SRR16526440, SRR16526439, SRR16526438, SRR16526437.). To do so, we calculated allele frequencies in each population, including *S. rodhaini,* individually. We then merged and filtered these frequency tables to retain only those absent in *S. rodhaini* (allele frequency = 0) and fixed in the respective focal population (allele frequency = 1).

### Allele frequency spectrum and empirical cumulative distribution function (ECDF)

We used the site.spectrum() function from the *pegas v1.2* R package [92] to compute the folded site frequency spectrum and bcftools v1.9 [89] to get overall allele frequency for SNPs in each individual population. We used stat_ecdf() from *ggplot* to calculate and plot ECDF for a statistical comparison of laboratory and field populations.

### Linkage disequilibrium

We examined linkage disequilibrium (LD) between autosomal variants within each population with PLINK v1.90b6.21 to make pairwise comparisons between SNPs within 1Mb of one another (--ld-window-r2 0.0, --ld-window 1000000, --ld-window-kb 1000). We binned average $r^2$ values using stats.bin() from the *fields* v14.1 R package [93] into 1,000 equal windows along the log scale which were calculated with logseq() from the *pracma v2.4.4* package [94]. Rare variants (MAF < 0.05) were excluded from this analysis. We fitted LD decay curves to these points using locally estimated scatter-plot smoothing (LOESS), and compared LD decay curves, we measured the distance at which LD is reduced to $r^2 = 0.5$ ($LD_{0.5}$). We used a custom script to calculate LD between unlinked markers. Briefly, we assigned genomic in lieu of chromosome positions to 10,000 randomly selected variants in each population and used PLINK to calculate LD (--ld-window-r2 0.0, --ld-window 999999, --ld-window-kb 10000). We then reassigned chromosomes and excluded results where R2 was calculated between markers on the same chromosome.

### Census and Ne estimation

Census: We estimated the number of adult genotypes each generation (census size) data using detailed schistosome life cycle maintenance records we keep for each of our laboratory populations. Specifically, we estimated the number of parasite genotypes present within shedding snails used to infect the hamster hosts. Generally, we infect individual snails with five to ten miracidia and record the number of infected and uninfected snails at the time of the first shedding. Therefore, the probability of a snail not being infected is:

$$P(0) = \frac{\textit{Number of uninfected snails}}{\textit{Number of surviving snails}}$$

We then computed the probabilities that each shedding snail contains 1,2…n parasite genotypes utilizing a Poisson distribution with the dpois() function from the *stats v4.2* package [95,96]. We note that this provides an upper limit on the number of adult worm genotypes within hamsters, because some cercarial genotypes may fail to establish.

$N_e$ estimation: We used two programs to determine effective population size: NeEstimator v2 [97], which relies on linkage disequilibrium between pairs of SNPs on different chromosomes to estimate $N_e$ and COLONY v2 [98], which calculates $N_e$ based on sibship inference. We used the R package *radiator* v1.2.8 [99] to convert working VCF files per population (14,073 – 119,643 loci) to suitable input files for each software and ran COLONY via the command line with default parameters. Additionally, we created an input file listing chromosomes and loci to run NeEstimator v2 with the "LD Locus Pairing" option which excludes the comparison of loci on the same chromosome.

## Bottleneck simulation

We used *vcfR* v1.13.0 [100] to extract genotypes from a VCF file containing common variants in the Brazilian field population. We then randomly sampled 10,000 loci to generate an input file suitable for BottleSim v2.6 [101]. We simulated bottleneck events with the "Diploid multilocus, constant population size" option, assuming no overlap between generations, and dioecy with random mating. We ran this simulation for $N = 400, 200, 100, 50, 25,$ and $5$ for 400 generations with a 1:1 sex ratio. This simulation approach is applicable to any sexually reproducing organism and examines the loss in heterozygosity (He) over time in parasite populations maintained with differing numbers of breeding adult parasites.

## Statistical analysis

We performed all statistical analyses with R package *rstatix* v0.7.2 [102] or *stats* v4.2. We used Student's t-tests (parametric) or Wilcoxon's rank-sum test (non-parametric) to compare the means of field and laboratory populations (normally distributed data, Shapiro test, $p > 0.05$). To compare empirical cumulative distributions (ECDFs) describing site frequency spectra, we used Kolmogorov-Smirnov statistic and permutation tests. We conducted pairwise comparisons between all populations examined (36 pairwise comparisons). We then calculated the mean K-S statistic for 16 comparisons of field and lab populations, and compared this to 10,000 randomly permuted datasets. We considered comparisons statistically significant when $p < 0.05$ [32].

## Supporting information

**S1 Fig.  PCA plot including field and laboratory populations.** PCA plot showing clusters of all populations used in this study.
(TIFF)

**S2 Fig.  Relationship between diversity and chromosomal location.** We calculated nucleotide diversity (π) in 25kb windows separately in each laboratory *S. mansoni* population. The distance from each window to the nearest chromosome end was calculated as a percentage of the total chromosome length and binned to the nearest whole percentage point. Box plots indicate the range of π values across all populations at each particular distance bin. There was no relationship between mean π and proximity to the chromosome ends, but variance in π increased at the chromosome ends. This result was also obtained with windows of 5 and 2 kb, so is robust to window size.
(PNG)

**S3 Fig.  Average number of nucleotide differences per site ($D_{XY}$) for all pairwise combinations Schistosome populations.** Data from all pairwise combinations are plotted on the same graph: red dots indicate comparisons between lab and field populations, green dots field vs field populations and blue dots are lab vs lab populations. The gene content in the $D_{XY}$ peaks are shown in the table below.
(PNG)

**S4 Fig.  Folded allele frequency spectra.** (A) Histograms of folded allele frequency spectra of each *S. mansoni* population. (B) Permutation tests to compare ECDFs from field and laboratory populations. We conducted pairwise comparisons between all populations examined (36 pairwise comparisons). We then calculated the mean K-S statistic for 16 comparisons of field and lab populations, and compared this to 10,000 randomly permuted datasets. The histogram shows the distribution of permutated values, with the empirical value and one tailed permutation test statistic marked by the red arrow.
(PNG)

**S5 Fig. LD decay between exonic SNPs in all *S. mansoni* populations.** (A) $r^2$ showing LD decay with physical distance between exonic SNPs along the chromosomes. Mean was calculated over 1 kb windows following the log scale except for SmBRE for which all data points were plotted. (B) Bar plot showing position when $r^2 = 0.5$ ($LD_{0.5}$) for field and laboratory populations. A *t*-test was used to compare field and laboratory populations.
(TIFF)

**S6 Fig. Unlinked LD.** Box and whisker plot showing LD (squared correlation coefficient, R2) of unlinked variants in each population.
(TIFF)

**S7 Fig. Estimated census size ($N_c$) of laboratory *S. mansoni* populations.** (A) Line plot showing estimated census size over time. We used detailed life cycle maintenance records to estimate P(0) and calculated numbers of parasites/ snail assuming a Poisson distribution. Note that these $N_c$ values are likely to be systematic overestimates. We conduct hamster infections with newly infected batches of snails to which we add surviving infected snails from the prior life cycle maintenance. Therefore, the proportion of uninfected snails (P(0)) will be underestimated, and Poisson estimates of numbers of parasite genotypes per snail will be overestimated. The actual $N_c$ values are likely to be somewhat lower. (B) Bar plot showing the harmonic mean of the $N_c$ for each population. The error bars represent a 95% confidence interval. (C) Scatter plot showing the relationship between $N_e$ as calculated by COLONY (filled circle) and NeEstimator (open circle) for each population. The lines represent a linear regression model, and the corresponding Pearson correlation coefficients are displayed in accordance with the legend of the tool used.
(PDF)

**S1 Table. Sample information.**
(XLSX)

**S2 Table. Nucleotide diversity in wild and domesticated/lab adapted animals and plants.**
(XLSX)

## Acknowledgments

Snails infected with SmNMRI parasites were provided by the Schistosomiasis Resource Center of the Biomedical Research Institute (Rockville, MD) through NIH-NIAID Contract HHSN272201700014I. We thank Sarah Schmid and Gabrielle Bate for conducting the monomiracidial snail infections and coordinating shipping and Dr. Margaret Mentink-Kane for her assistance.

## Author contributions

**Conceptualization:** Kathrin S. Jutzeler, Timothy J.C. Anderson.

**Data curation:** Kathrin S. Jutzeler, Julie Dunning Hotopp.

**Formal analysis:** Kathrin S. Jutzeler, Roy N Platt, Julie Dunning Hotopp.

**Funding acquisition:** Timothy J.C. Anderson.

**Investigation:** Kathrin S. Jutzeler, Robbie Diaz, Madison Morales, Winka Le Clec'h.

**Methodology:** Roy N Platt, Winka Le Clec'h, Frédéric D. Chevalier.

**Project administration:** Timothy J.C. Anderson.

**Resources:** Frédéric D. Chevalier.

**Supervision:** Roy N Platt, Winka Le Clec'h, Frédéric D. Chevalier, Timothy J.C. Anderson.

**Visualization:** Kathrin S. Jutzeler, Roy N Platt.

**Writing – original draft:** Kathrin S. Jutzeler, Timothy J.C. Anderson.

**Writing – review & editing:** Kathrin S. Jutzeler, Roy N Platt, Winka Le Clec'h, Frédéric D. Chevalier, Timothy J.C. Anderson.

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
