## [Decision Letter · Decision Letter 0]

13 Dec 2024

PPATHOGENS-D-24-02401

Abundant genetic variation is retained in many laboratory schistosome populations

PLOS Pathogens

Dear Dr. Anderson,

Thank you for submitting your manuscript to PLOS Pathogens. After careful consideration, we feel that it has merit but does not fully meet PLOS Pathogens's publication criteria as it currently stands. Therefore, we invite you to submit a revised version of the manuscript that addresses the points raised during the review process.

Please submit your revised manuscript within 30 days Feb 11 2025 11:59PM. If you will need more time than this to complete your revisions, please reply to this message or contact the journal office at plospathogens@plos.org. Please include the following items when submitting your revised manuscript:

We look forward to receiving your revised manuscript.

Kind regards,

Erik C. Andersen, Ph.D.

Guest Editor

PLOS Pathogens

James Collins III

Section Editor

PLOS Pathogens

Sumita Bhaduri-McIntosh

Editor-in-Chief

PLOS Pathogens

orcid.org/0000-0003-2946-9497

Michael Malim

Editor-in-Chief

PLOS Pathogens

orcid.org/0000-0002-7699-2064

**Journal Requirements:**

At this stage, the following Authors/Authors require contributions: Kathrin S. Jutzeler, Roy N Platt, Robbie Diaz, Madison Morales, Winka Le Clec'h, Frédéric Chevalier, and Timothy JC Anderson. Please ensure that the full contributions of each author are acknowledged in the "Add/Edit/Remove Authors" section of our submission form.

- ® on Line: 374.

Potential Copyright Issues:

- Figures 1 and 2; Please confirm whether you drew the images / clip-art within the figure panels by hand. If you did not draw the images, please provide a link to the source of the images or icons and their license / terms of use; or written permission from the copyright holder to publish the images or icons under our CC BY 4.0 license. Alternatively, you may replace the images with open source alternatives. See these open source resources you may use to replace images / clip-art:

6) Please ensure that the funders and grant numbers match between the Financial Disclosure field and the Funding Information tab in your submission form. Note that the funders must be provided in the same order in both places as well. State what role the funders took in the study. If the funders had no role in your study, please state: "The funders had no role in study design, data collection and analysis, decision to publish, or preparation of the manuscript.".

**Reviewers' Comments:**

Reviewer's Responses to Questions

**Part I - Summary**

Reviewer #1: See attached

Reviewer #2: See attachment

Reviewer #3: The manuscript by Jutzeler and colleagues describes their discovery of high levels of genetic diversity in schistosome populations maintained in the laboratory. This is an important and relevant finding, with bearing on the use of laboratory helminth populations to study the genetic basis of traits such as drug resistance. The experimental design seems robust, the genomic analyses and simulations seem appropriate, and the data support the claims. The paper is clearly written and was enjoyable to read, although I felt in a couple of places there was an expectation that the reader was already familiar with the papers cited; please always state the key point briefly rather than simply providing a reference for the reader to look up. I have no major corrections but have several minor edits or comments/queries.

**Part II – Major Issues: Key Experiments Required for Acceptance**

Reviewer #1: See attached

Reviewer #2: (No Response)

Reviewer #3: None

**Part III – Minor Issues: Editorial and Data Presentation Modifications**

Reviewer #1: See attached

Reviewer #2: (No Response)

Reviewer #3: 72. Please state the country the SmNMRI population was isolated from.

Fig 1. Panel on RHS would be improved by showing males and females within each pair in different colours.

Fig 3. Did you also generate a PCA with the exonic data only to allow inclusion of field samples?

144. The 51% reduction presumably reflects the very high diversity of the Tanzanian field population (and the low diversity of SmBRE), but as far as is stated in the paper, none of the lab isolates originated from Tanzania. What is the reduction if you do the most relevant pairwise comparisons e.g. each Brazilian lab isolate with the Brazilian field sample?

148. Did you look at which genes were encoded in regions of high nucleotide diversity across the different isolates? In Discussion, suggest that balancing selection due to host immune response could maintain genetic variation as per Stevens et al 2023, so might expect to see enrichment for gene families that interact with the host immune response in these loci?

169. I don’t think ‘nearly significantly’ is appropriate description.

193. ‘The reasons for this are explained elsewhere [19].’ Please state the reasons briefly.

199. Are overlapping generations relevant for laboratory maintenance of schistosome populations?

256. Worth also commenting on the distinct allele frequency spectra here (Fig S1).

Table 4. Could be interesting to include data from lab and field isolates of other well studied outbreeding helminth populations (e.g. C. remanei and H. contortus) if available for comparison.

349. Please add ‘or PacBio’

373. Individual males and female heads were sequenced for SmBRE only, but previously stated (line 363) the avoidance of sampling adults in case derived from clonal cercariae from a single snail – is there a risk this did happen for SmBRE and relates to the very low genetic variation measured in this isolate? Good to highlight any technical differences in the discussion.

375 and 384. Do you expect any bias from the WGA step that could affect the results?

414. Unclear why lifted merged vcf to different genome assembly?

PLOS authors have the option to publish the peer review history of their article (what does this mean? ). If published, this will include your full peer review and any attached files.

**Do you want your identity to be public for this peer review?** For information about this choice, including consent withdrawal, please see our Privacy Policy .

Reviewer #1: No

Reviewer #2: **Yes: ** Stephen R. Doyle

Reviewer #3: No

**Figure resubmission:**
---

## [Decision Letter · Decision Letter 1]

8 Aug 2025

Dear Dr. Anderson,

We are pleased to inform you that your manuscript 'Abundant genetic variation is retained in many laboratory schistosome populations' has been provisionally accepted for publication in PLOS Pathogens.

Best regards,

Erik C. Andersen, Ph.D.

Guest Editor

PLOS Pathogens

James Collins III

Section Editor

PLOS Pathogens

Sumita Bhaduri-McIntosh

Editor-in-Chief

PLOS Pathogens

orcid.org/0000-0003-2946-9497

Michael Malim

Editor-in-Chief

PLOS Pathogens

orcid.org/0000-0002-7699-2064

Reviewer Comments (if any, and for reference):

Reviewer's Responses to Questions

**Part I - Summary**

Reviewer #2: Thanks to the authors for their efforts and responses. All looks good to me. I look forward to seeing it in print.

Many kind regards,

Stephen Doyle

Reviewer #3: I am satisfied that the authors have addressed all my concerns in their revised submission.

**Part II – Major Issues: Key Experiments Required for Acceptance**

Reviewer #2: (No Response)

Reviewer #3: (No Response)

**Part III – Minor Issues: Editorial and Data Presentation Modifications**

Reviewer #2: (No Response)

Reviewer #3: (No Response)

PLOS authors have the option to publish the peer review history of their article (what does this mean? ). If published, this will include your full peer review and any attached files.

**Do you want your identity to be public for this peer review?** For information about this choice, including consent withdrawal, please see our Privacy Policy .

Reviewer #2: **Yes: ** Stephen R. Doyle

Reviewer #3: No

---

## [Editor Report · Acceptance letter]

Dear Dr. Anderson,

We are delighted to inform you that your manuscript, "Abundant genetic variation is retained in many laboratory schistosome populations," has been formally accepted for publication in PLOS Pathogens.

Best regards,

Sumita Bhaduri-McIntosh

Editor-in-Chief

PLOS Pathogens

orcid.org/0000-0003-2946-9497

Michael Malim

Editor-in-Chief

PLOS Pathogens

orcid.org/0000-0002-7699-2064